# Identification of Potential Harmful Transformation Products of Selected Micropollutants in Outdoor and Indoor Swimming Pool Water

**DOI:** 10.3390/ijerph19095660

**Published:** 2022-05-06

**Authors:** Edyta Kudlek, Anna Lempart-Rapacewicz, Mariusz Dudziak

**Affiliations:** Faculty of Energy and Environmental Engineering, Silesian University of Technology, Konarskiego 18, 44-100 Gliwice, Poland; anna.lempart-rapacewicz@polsl.pl (A.L.-R.); mariusz.dudziak@polsl.pl (M.D.)

**Keywords:** micropollutants, swimming pools, decomposition, by-products

## Abstract

This paper presents the estimation of micropollutant decomposition effectiveness and the identification of transformation intermediates formed during selected processes used in the treatment of swimming pool water. Tests were carried out under both indoor and outdoor conditions to simulate the removal of contaminants in different types of pool water basins. Model swimming pool water spiked with caffeine, carbamazepine, bisphenol A and oxadiazon were subjected to chlorination, ozonation, UV radiation, and artificial and sun lightening, carried out as single or combined processes. It was noted that organic micropollutants decompose faster during exposure to natural sunlight than artificial lighting. Caffeine and carbamazepine belong to compounds that are resistant to single ozone or light decomposition. Bisphenol A was completely removed by the action of the chlorination agent NaOCl. The highest compound removal degrees were noted for the integrated action of natural sunlight, NaOCl and O_3_. This process allows also for the decomposition of all caffeine and oxadiazon decomposition by-products that potentially are toxic to swimming pool users.

## 1. Introduction

The presence of thousands of different inorganic and organic micropollutants in the whole water environment becomes one of the most current and important issues in environmental engineering. The occurrence of these potentially harmful-to-swimmer-health compounds in swimming pool water has been confirmed in numerous scientific articles [1,2,3]. Fill water and swimmers are considered to be the biggest source of micropollutants in this special water environment [4]. Micropollutants may be transformed or decomposed under the influence of many factors, such as disinfection agents, natural sunlight and artificial lighting [5]. Chlorine is widely used, not only in drinking water disinfectants [6] but also as an effective disinfection agent for swimming pools and recreational water [7].

Van Veldhoven et al. [8] pointed out that numerous molecular changes are a result of swimming in chlorinated pools. These changes can result from both the direct contact with the remaining chlorine and/or with the formed disinfection by-products. Richardson et al., [9] reported on more than 600 disinfection by-products that can occur in this specific environment. However, this number still does not include all the chlorination intermediates detected in different types of disinfected recreational water reservoirs. Numerous studies are focused on the well-known disinfection by-products, such as chloramines, trihalomethanes (THMs), haloacetic acids (HAAs), chloral hydrates (CH), haloketones (HKs), dichloromethylamine (CH_3_NCl_2_), cyanogen chloride (CNCl), haloacetonitriles (HANs) and nitrosamines detected in swimming pool water [9,10,11,12], however, the decomposition of the by-products of contaminants coming from the surrounding pool space and the pool users themselves are considered a significant risk to swimmers. Lempart et al. [13] identified more than 100 contaminates, including medicines, hormones, vitamins, industrial admixtures or cosmetic ingredients. Chlorine can cause the decomposition and transformation of these compounds [14]. Gibs et al. [15] and Nam et al. [16] noted that the chlorination process used in water treatment plants leads to the removal range from 6% to 100% of several compounds. Higher chlorine doses applied to swimming pool water can therefore affect a more effective decomposition of micropollutants and also lead to the formation of a larger number of intermediates.

According to the data from the International Agency for Research on Cancer (IARC), the same intermediates are considered to be possibly carcinogenic to humans [17]. Therefore, there is a need to examine the reactions occurring between the used disinfection agent and the impurities in the pool water. The obtained knowledge may lead to the development of methods for the protection of pool users against harmful compounds without the generation of a wide range of toxic intermediates. In this matter, for example, membrane processes are considered [18].

The performed research was focused on the examination of the possible transformation pathways of selected micropollutants from the groups of pharmaceutical and personal care products (carbamazepine, caffeine), industrial additives (bisphenol A) and pesticides (oxadiazon) in swimming pool water. The compounds were selected based on the preliminary studies that focused on the determination of compounds in different swimming pools located in south Poland [3,13,19]. Meffe and de Bustamante [20] report the occurrence of many types of pesticides in surface water and groundwater, which is a source of tap water. Carbamazepine, however, was also the most frequently identified pharmaceutical compound in swimming pools in the study conducted by Ekowati et al. [21], whereas Suppes et al. [4] identified this compound in 16 out of 32 studied swimming pools in Minnesota and Wisconsin; tap water is one of the most common organic micropollutant sources in this environment. Model swimming pool water constituted the research subject, which was subjected to chlorination, ozonation and UV radiation. These tests were also carried out in both outdoor and indoor conditions to investigate the differences between the decomposition of compounds exposed to natural sunlight and artificial lighting.

## 2. Materials and Methods

### 2.1. Tested Samples

The research was conducted on two types of model solutions: a tap water solution and model swimming pool water prepared based on tap water chlorinated with the use of the aqueous solution of sodium hypochlorite (NaClO). The chlorination agent NaClO was used as a 15% solution and was purchased from Chemoform, Poland. The total chlorine concentration in the prepared water samples was sustained at 1.0 mg·L^−1^. The prepared water samples were spiked with organic micropollutants, which belong to the group of contaminants of emerging concern (CECs): pharmaceutical and personal care products—caffeine (CAF) and carbamazepine (CBZ); industrial additives—bisphenol A (BPA); and pesticides—oxadiazon (ODZ). The initial concentration of the compounds was set at 200 µg·L^−1^. The high concentration, which significantly exceeded the real concentration of those compounds identified in real swimming poll water samples [3,18] was used to facilitate the analytical procedure and to allow for the identification of formed by-products. It should be noted that the concentration of by-products in experiments performed on real swimming pool water can be placed below the limit of detection of the used equipment. The parameters of the model swimming pool were given in Table 1. The pH, conductivity and temperature were measured by the use of the multifunctional meter CX-461 from ELMETRON (Zabrze, Poland). The total organic carbon (TOC), inorganic carbon (IC) and total carbon (TC) were measured with the TOC-L analyzer by Shimadzu (Kyoto, Japan), and the free chlorine, combined chlorine and total chlorine were measured in accordance with the cuvette tests for the Spectroquant^®^Pharo 300 photometer by Merck Sp. z oo. (Warszawa, Poland).

### 2.2. Chlorination and Ozonation Processes

The test was devoted to the evaluation of the influence of the NaOCl dose on the removal degree of CECs and performed for the tap water solution (without previous chlorination). Experiments were conducted in a laboratory glass reactor with a volume of 1.0 L placed on magnetic stirrers. The concentrations of the total chlorine chosen for this experiment ranged from 0.5 to 1.0 mg·L^−1^. The reaction time was set at 5, 10 and 15 min. The same reaction times were selected for the process of the ozonation of tap water solutions. Further, O_3_ was introduced in the model swimming pool water samples by the use of a ceramic diffuser from an O_3_ generator, the Ozoner FM500 WRC Multiozon (Sopot, Poland). The concentrations of ozone were equal to 1.0, 1.2, 1.4, 1.6, 1.8 and 2.0 mg·L^−1^. They were measured at the inlet to the reactor by the use of the photometric method O_3_ with the Spectroquant^®^ by Merc Sp. z oo. (Warszawa, Poland). The ozonation reaction was stopped with the introduction of the post-processed samples of 24 mmol∙L^−1^ of Na_2_SO_3_. In order to omit the influence of light on the decomposition of the tested compounds, all experiments were conducted in a dark chamber. Experiments for all tested compounds were carried out separately.

### 2.3. Photodecomposition Processes

Tap water and swimming pool water solutions were subjected to natural sunlight (outdoor experiment) and artificial light (indoor experiment). The outdoor experiment was conducted in laboratory borosilicate beakers with a volume of 1.0 L during summer. Beakers were constantly stirred to ensure the proper mixing of the entire reacting volume. The temperature of the tested solutions was continuously measured during the experiments by the use of the multifunctional meter CX-461 from ELMETRON (Zabrze, Poland) and ranged from 24.0 to 26.0 °C. The reaction times were equal to 30, 60 and 300 min. The indoor experiment was performed in the same laboratory beakers exposed to light, which was emitted by 58 W light bulbs with a luminous flux of 5000 Lumen.

The outdoor and indoor experiment was repeated for swimming pool water solutions (with a total chlorine concentration of 1.0 mg·L^−1^), which were additionally subjected to UV radiation and ozonation. The UV light was emitted by Heraeus (Hanau, Germany), which emitted radiation with a wavelength of λ = 254 nm. The radiation flux, according to the producer of the lamp, was equal to 1.667 J·s^−1^·m^−2^. The temperature of the reaction mixture was 23 ± 1 °C and the irradiation time was set at 30, 60 and 300 min. O_3_ was applied in a dose of 2.0 mg·L^−1^. The photodecomposition processes were also carried out separately for all the tested compounds.

### 2.4. Compound Analytical Procedure

The concentration of the tested CECs in the initial and water samples, after the implementation of disinfection methods, was estimated using gas chromatography with mass detection (GC-MS) equipped with electron ionization. The compounds were extracted from the tested samples by solid-phase extraction (SPE).

The pH of the samples before SPE was adjusted with the use of 0.1 mol∙L^−1^ NaOH (purity grade > 99.8%) to 7.0. The extracted volume of the water samples was equal to 20 mL. The Supelclean™ ENVI-18 by Sigma-Aldrich (Poznań, Poland) was used as an extraction column. The column bed of those types of cartridges was composed of silica gel base material with polymericallyoctadecyl bonding (bed weight of 1000 mg). The bed pore size was equal to 60 Å and the total bed surface area reached 475 m^2^ g^−1^. The conditioning of the cartridges was performed with the use of 5.0 mL of ACN and the same volume of methanol, whereas the cartridge washing before sample extraction was implemented using 5.0 mL of deionized water (pH equal to 7.0). The extracts were eluted with 1.5 mL of methanol and 1.5 mL of ACN, respectively. The recovery of the tested compounds using this procedure exceeded 99%.

The extract was subjected to the chromatographic analysis performed by the 7890B GC-MS(EI) chromatograph by Perlan Technologies (Warszawa, Poland). The chromatograph was equipped with a silphenylene polymer capillary column, SLB^®^-5 ms 30 m × 0.25 mm of 0.25 μm film thickness, from Sigma-Aldrich (Poznań, Poland). The oven temperature program and other temperature details were given in previous self-studies [22].

To ensure both low detection limits of the analysed CECs and the identification of the formed decomposition by-products, all extracts were analyzed twice, in the selected ion monitoring mode (SIM) and in the total ion current mode (TIC).

The percentage of removal of each tested CEC after the subjection to disinfection methods and natural and artificial UV lighting was calculated according to Equation (1):(1)$$Removal\;\left(\%\right)=\frac{C_{i}-C_{p}}{C_{i}}\cdot 100,$$
where *C_i_* and *C_p_* are the initial and post-processed compound concentrations (mg·L^−1^), respectively.

### 2.5. Evaluation of Research Results

The study analyzes the efficiency of the removal of four selected compounds during three processes taking place in the swimming pool installations (chlorination, ozonation and photodegradation). This is expressed as the removal degree [%] depending on the characteristic parameters of the individual process (duration time of the process, the chlorine concentration for chlorination or ozone dose during ozonation). The description of the parameters and the determination of the relationship between them were carried out using a Microsoft Excel spreadsheet.

The main goal of the statistical analysis was to assess the significance of the dependence of removal degrees on the parameters of the individual process. Initial data analysis (comparison of distributions and assessment of their similarity) indicated the necessity to use non-parametric tests for the evaluation of the test results. An ANOVA of the rank of Kruskal–Wallis was used for this purpose.

A significance level α = 0.05 was adopted for the calculations. As a preliminary assumption, no dependence was adopted when the test probability was greater than the significance level (*p* ≥ α). The occurrence of statistically significant differences between the analyzed parameters was found when the test probability was lower or much lower than the adopted level of significance (*p* < α or *p* << α).

For the graphical presentation and comparison of the measurement results, bar charts were used to provide a general presentation of the ordered and grouped test results. Each water sample was analyzed in triplicate, and the presented results are the mean values of these replicates.

The marked assignment errors were estimated on the basis of the standard deviation for three repetitions of each test. The standard deviation values for all tested samples did not exceed 2.5%, which indicates the high reproducibility of the obtained results.

## 3. Results

### 3.1. Decomposition of Compounds during the Process of Chlorination and Ozonation

The first stage of the conducted experiments was dedicated to the determination of the removal degrees of each tested CEC. Tap water solutions of the compounds were subjected to six different doses of NaOCl. The whole experiment was conducted in a dark chamber in order to eliminate the influence of light on the decomposition of compounds. The total chlorine concentration measured after adding the chlorinating agent was equal to 0.5, 0.6, 0.7, 0.8, 0.9 and 1.0 mg·L^−1^. The chlorination experiment was conducted for 5 min and the removal degrees of tested CECs are presented in Figure 1.

The highest removal degree was noted for BPA. It exceeds 99% for the lowest concentration of total chlorine. In the case of this compound, the differences in the removal degree achieved for different total chlorine concentrations were not significant (*p* ≥ 0.05). The removal degrees, noted for the lowest chlorine concentration for other tested compounds, reached only 14% for CAF, 10% for CBZ and 9% for ODZ. An increase in the dose of total chlorine significantly affected the removal degree (*p* < 0.05). However, the increase in process efficiency was slight. The concentration of CAF and CBZ after contact with 1.0 mg·L^−1^ of total chlorine decreased by 26%. The removal of ODZ after the application of the highest dose of the chlorination agent ranged at 13% and it did not change with the increasing time of the chlorination process (Figure 2) (*p* ≥ 0.05).

The influence of the chlorination time on the removal degree of CECs was examined in the next stage of the study (Figure 2). In general, the main reactions between the chlorination agent and the tested CECs-occurred during the first 5 min of the process. The removal degrees noted after longer process times of 10 or 15 were less than 1% higher for CAF and CBZ than those noted after a 5 min exposure to the chlorination agent. The influence of time on the removal degree of all tested compounds was not significant (*p* ≥ 0.05).

Similar decomposition experiments were conducted for CEC water solutions subjected to the process of ozonation. The contact time was set for 15 min. The micropollutant removal degrees noted for O_3_ doses ranging from 1.0 to 2.0 mg·L^−1^ are presented in Figure 3. CAF is considered to be resistant to the influence of O_3_. The concentration of this CEC in post-processed water samples was the same as in the initial solutions. For CBZ, BPA and ODZ, the ozonation time significantly affected the removal degree (*p* < 0.05).

These results correspond to studies conducted on deionized water solution, where those two compounds were also characterized by a negligible removal [5]. The BPA and ODZ concentration decreased with the increase of the O_3_ dose and reached 55% and 16% (O_3_ dose equal to 2.0 mg·L^−1^), respectively. The contact time between the tested CECs solutions and the O_3_ had a notable impact only in the case of BPA.

Figure 4 presents the changes in the compound concertation subjected to 2.0 mg·L^−1^ of O_3_ after 5, 10 and 15 min of process time. The removal degrees of CAF and ODZ observed at all time periods were similar. Only the concentration of BPA decreased with the increase of the processing time and ranged from 45% noted after 5 min of ozonation to 55% after 15 min. For BPA, the influence of ozonation time was significant for the results of removal efficiency (*p* < 0.05). For other compounds, it was not significant (*p* ≥ 0.05).

### 3.2. Photodecomposition of Compounds in Outdoor and Indoor Conditions

The next stage of the experiment was focused on the influence of the lightening of the water on the self-decomposition of compounds. Therefore, tap water CECs solution was subjected to artificial lighting, which imitated the conditions occurring in an indoor swimming pool, and sun lightening to simulate outdoor swimming poll conditions. The experiment was performed without the addition of chlorinating agents to the solutions, and it shows the significant effect (*p* < 0.05) of the photodecomposition process on the removal degree of the tested compounds. Figure 5 shows the concentrations decrease of all the micropollutants tested at 30, 60 and 300 min of exposure to natural sunlight and artificial lighting. Natural sunlight leads to faster decomposition of the tested micropollutants than artificial lighting. Therefore, it can be concluded that the half-life of the organic compound in outdoor pools is shorter than in indoor pools. For example, the removal rate noted for CBZ in the outdoor experiment ranged from 3% (30 min of sun lightening) to 9% (300 min of sun lightening), while during the indoor experiment the concentration decreased by 0.2% (30 min exposure to artificial light) to 5% after 300 min of artificial irradiation.

After the irradiation of the tested mixtures with an additional source of light in the form of UV light, which is often used in swimming pool water treatment technologies, the removal degree of BPA and ODZ increases. Moreover, the implementation of O_3_ increases the decomposition of all tested micropollutants. More notable changes in the concentration were observed during the irradiation of the ODZ solutions. This compound decomposed by over 55% after 30 min of sunlight exposure, and after 300 min, its removal degree reached 85%, while the decrease in ODZ concentration after 300 min of artificial lightening reached only 35%. The obtained results were significantly affected by the time of the photodecomposition process (*p* < 0.05).

Figure 6 presents the results obtained during the exposition of chlorinated micropollutant tap water mixtures to artificial light and sunlight. The presence of NaOCl significantly improved the decomposition of all tested CECs. The compounds were simultaneously decomposed by the action of the chlorination agent and the photodecomposition reactions. Complete removal of BPA was noted in samples subjected to indoor and outdoor conditions. The removal degree of ODZ during sunlight irradiation, supported by the presence of chlorine, increased to 88%. The presence of chlorine also leads to the increase of ODZ decomposition during artificial lightning. The removal degree increased from 35%, achieved without chlorine to 50%. However, the removal degrees of CAF and CBZ in both lightening methods exceeded 27%.

To improve the decomposition processes of the compounds, the chlorinated and irradiated water mixtures were additionally subjected to UV irradiation from a second source with a wavelength of λ = 254 nm (Figure 7). UV irradiation is a commonly used method of swimming pool water treatment that supports the action of chlorination methods [23]. For BPA and ODZ in both outdoor and indoor conditions, the results of removal degrees significantly depend on the time of the process (*p* < 0.05), while for CAF, both were not significant (*p* ≥ 0.05). In the case of CZB, for outdoor swimming pools (*p* < 0.05), the results were significant (Figure 7a) and not significant (*p* ≥ 0.05) for indoors (Figure 7b).

With the increase in the simultaneous irradiation of UV light and artificial or natural sunlight, an increase in the compound removal degrees was observed. This indicates a positive impact of the UV disinfection method on the decomposition of compounds. For example, the removal rates of CAF and CBZ after 300 min of the experiment duration in outdoor conditions exceeded 33% and 35%, respectively, while in indoor conditions the CAF concentration was reduced by 31% and the CBZ concentration decreased by nearly 30%. A significant improvement in the removal degree in indoor conditions was also noted in the case of ODZ. The removal degree after 30 min of simultaneous UV and artificial irradiation exceeded 62% and increased with the processing time to over 86% (300 min of process elongation).

The influence of the simultaneous action of NaOCl, O_3_ and artificial or sunlight irradiation were also examined. Figure 8 presents the results obtained in outdoor and indoor conditions. The presence of O_3_, which besides its oxidation properties, is also an additional source of reactive species generated under the action of sunlight, leads to an increase of all compound removal degrees during the outdoor experiment. For example, the BPA concentration was reduced after 30 min by over 83% and after 300 min, a complete removal of this compound was noted. An inverse relation was observed during the indoor experiment. The presence of O_3_ leads to the inhibition of the compound decomposition. Therefore, it can be supposed that only sunlight and artificial UV light are able to generate reactive species that can react with the contaminants occurring in swimming pool water. The impact of artificial UV light and the O_3_ effect, also on the by-products, formed during the implemented processes.

### 3.3. Identification of the Decomposition Intermediates

The TIC chromatographic analysis allowed for a more specific analysis of the post-processed samples. The obtained chromatograms realize, in addition to the peaks corresponding to parent compounds, several other peaks with lower signal intensity. These peaks corresponded to decomposition intermediates formed during the implemented processes. The comparison of the mass spectra of the newly formed compounds with the NIST v17 database software allowed for their identification. Table 2 summarized all the intermediates, which were identified with an over 85% similarity to the database patterns. Three compounds could not be identified by the database. The signal intensity of those compounds was similar to the signals recorded for the identified compounds. Therefore, their mass spectra were analyzed based on literature data.

The intermediates were mostly identified during the first 30 min of process duration. The signal intensity increased in the time from 0 to 30 min, and then between 30 and 60 min, the decrease in signal intensity was noted, which may indicate a gradual decomposition of these compounds. In addition, no presence of these compounds was recorded after 300 min (especially in the case of the post-processed water samples of NL/Cl_2_/UV, AL/Cl_2_/UV, NL/Cl_2_/O_3_, AL/Cl_2_/O_3_).

## 4. Discussion of Results

Understanding the basic mechanisms of decomposition of compounds occurring in the swimming pool water was possible thanks to a series of experimental tests carried out on model water solutions subjected to individual physical or chemical processes.

The effectiveness of the decomposition of compounds during the process of chlorination (Figure 1) and ozonation (Figure 3) depends on the dose of the oxidant. These processes are most intense in the first 5 min after dispensing (Figure 2 and Figure 4). However, not all compounds show susceptibility to the action of a single unit process, which results from the different structures of individual molecules and different physicochemical properties. CAF was resistant to ozonation and CBZ was only slightly removed during this process (Figure 3), which was also observed by Soufan et al. [24] and in previous studies conducted by Gibs et al. [15]. However, Weng et al. [25] noted a 72% removal of CBZ during breakpoint chlorination under neutral conditions. Furthermore, the physicochemical properties of compounds determine also their susceptibility to decomposition under the influence of different reactants. The process of chlorination (Figure 1) was more effective in the decomposition of CAF, CBZ and BPA, while higher removal degrees of ODZ were noted in samples subjected to the ozonation process (Figure 3).

While for the ozonation and chlorination processes, time was not significant, except for the BPA ozonation process (Figure 2 and Figure 4), in the case of unitary photodegradation processes, both under the influence of sunlight (Figure 5a) and artificial (Figure 5b). This parameter was of significant importance.

This dependency changed when chlorinated water was exposed to sunlight (Figure 6a) or artificial light (Figure 6b). For both, sunlight and artificial light, the duration of the process was significant only for the removal degree of ODZ (Figure 6). Among the tested four compounds, ODZ was the least susceptible to decomposition under the influence of reactive chlorine species (Figure 1). In the case of combining the chlorination and photodegradation processes (Figure 6), the degradation efficiency of this compound increased significantly over time.

The use of an additional source of radiation in the form of a UV lamp (Figure 7)—a commonly used device in swimming pool water disinfection technology [23]—increased the efficiency of CAF, CBZ and ODZ removal in relation to samples not exposed to the UV lamp.

In the case of BPA (which was almost 99% removed under the influence of chlorine—Figure 1), in the processes of ozonation (Figure 7) and irradiation of chlorinated water with a UV lamp (Figure 8), inhibition of decomposition was observed as compared to the process carried out without the participation of these two factors (Figure 6). Furthermore, it should be noted that the molar absorption coefficients of chlorine reactive species, such as HOCl, OCl^−^ at 254 nm (the emission peak of the UV lamp used in the study), are 59 M^−1^ cm^−1^ and 66 M^−1^ cm^−1^, respectively [26]. They are capable of absorbing and attenuating light at this given wavelength. The decomposition of compounds during the simultaneous action of chlorine reactive species and UV light can also be inhibited by the presence of low concentration levels of chloride ions Cl-. Those ions can come into reaction with other reactive chlorine species or hydroxyl radicals and form radicals with lower reduction potentials than HOCl or HO^•^ [27]. On the other hand, HO^•^ chlorine atoms (Cl^•^) play the main role during the decomposition of compounds in the UV/chlorination processes. These radicals are produced due to the UV photolysis of HOCl and OCl^−^ [28]; however, Wang et al. [29] and Nowell et al. [30] pointed out that HO^•^ acts as the major reactive species in the degradation of UV- and chlorine-persistent micro-pollutants.

Compounds during the irradiation with different light sources undergo direct or indirect photodegradation. Sunlight, as a mixture of UV-C, UV-B, UV-A, visible light and infrared, had the best potential to decompose compounds. Zhu et al. [31] reported that UV-C irradiation next to disinfection properties can also be used as an effective CECs elimination method.

The presence of organic matter and inorganic ions could positively or negatively affect the photodecomposition of compounds [32]. Moreover, Fang et al. [33] report that chlorine reactive species, in particular Cl^•^, react more quickly with acetic acid, benzoic acid and phenol than HO^•^. This suggests that functional groups of micropollutants based on these compounds will react faster with Cl^•^ than with HO^•^.

Figure 9, Figure 10, Figure 11 and Figure 12 present the possible transformation pathways of the tested micropollutants. The compounds, which were not clearly identified, were marked by a dotted line.

Samples containing CAF and ODZ are characterized by the lowest number of intermediates. During the reaction of CAF with NaOCl, 8-Chlorocaffeine was generated as a result of the aromatic electrophilic substitution at C-8 (Figure 9). Zarrelli et al. [34], with the use of high-performance liquid chromatography (HPLC) with a UV-Vis detector, also detected this CAF intermediate and other chlorination by-products: 1,3-Dimethyl-5- azabarbituric acid, N,N′-dimethyloxalamide and N,N′-dimethylparabanic acid. CAF samples subjected to the action of O_3_ did not contain any intermediate, which confirms the non-decomposition of these CECs (Figure 3), while the irradiation of the CAF water samples with artificial and natural sunlight leads to the formation of 8-Methoxycaffeine and 8-Hydroxymethylcaffeine. Those intermediates were also detected together with 8-Chlorocaffeine in samples irradiated in the presence of chlorine, whereas, the irradiation of chlorinated water samples with natural sunlight supported by UV or O_3_ allowed for complete decomposition of the CAF intermediates.

The analysis of the chromatograms obtained during the testing of the CBZ water samples subjected to the action of NaOCl indicated the formation of one compound, which could not be clearly identified by the NIST v17 software. This compound is characterized by the heaviest ion with an m/z value of 227, which is considered to be a molecular ion. In addition, two peaks in the molecular ion region were detected.

This indicated the presence of a Cl in the chemical structure of the compound. It can be supposed that this compound was generated during the chlorination of Iminostilbene (Figure 10). The Cl atom attached to the phenol ring and 3-Chloro-5H-dibenz[b,f]azepine was generated, which after the connection with a second Cl atom transforms into 3,7-dichloro-5h-dibenz[b,f]azepine. This possibility was also noted in previous studies focused on the chlorination of CBZ. Tandarić et al. [35] pointed out that Iminostilbene can undergo different transformation reactions promoted by reactive chlorine species. However, the two most feasible processes are the additions of the -Cl or -OH group on the C=C bond of this compound.

The presence of NaOCl leads also to the formation of 3-Hydroxycarbamazepine, 10,11-Dihydro-10-hydroxycarbamazepine, Dihydrocarbamazepine-10,11-trans-diol and Carbamazepine-10,11-epoxide. Those compounds are formed during the attack of oxidants on the compound molecule and the attachment to the HO^−^ group to the compound structure. Therefore, it can be concluded that they are always formed during the reactions caused by HO^•^ radicals, which are considered to be one of the most reactive and nonselective oxidation agents. This was confirmed by the presence of compounds with a hydroxyl group together with 9-Acridone and Acridine in the samples subjected to natural sunlight. The highest number of CBZ intermediates was noted for the samples exposed to natural sunlight supported by NaOCl and UV irradiation. Only two by-products were noted during the process of ozonation and artificial lightening. CBZ formed under the influence of O_3_ 3-Hydroxycarbamazepine and Carbamazepine-10,11-epoxide, while the irradiation with artificial light led to the formation of 3-Hydroxycarbamazepine and 10,11-Dihydro-10-hydroxycarbamazepine.

In general, the disinfection of swimming pool water by the use of NaOCl led to the formation of several chlorinated intermediates: HOCl (which is the major chlorinating species), OCl^−^, ClOH^•−^, Cl^•^ and Cl_2_^•−^. Tandarić et al. [35] studied the transformations of CBZ induced by the presence of HOCl and confirmed the generation of 10-Oxoiminostilbene, Carbamazepine 10,11-chlorohydrin, Carbamazepine N-chloramide, Dihydrocarbamazepine-10,11-trans-diol and Iminostilbene, which were observed in earlier studies performed by Soufan et al. [24]. The two latter compounds were also observed in this study.

The highest number of by-products was noted for samples containing BPA. Thirteen intermediates of this micropollutant were detected in samples subjected to natural sunlight irradiation, supported by the presence of NaOCl and samples exposed to artificial light in the presence of NaOCl and O_3_. As in the case of the identification of CBZ intermediates, two BPA by-products could not be identified based on the used mass spectral database. Those compounds were characterized by ions with an m/z value of 244 and 262, which are considered to be molecular ions. Therefore, according to the literature [36], it can be supposed that the observed peaks correspond to 5-Hydroxybisphenol and 5,5′-Dihydroxybisphenol, respectively (compounds marked in Figure 11 with a dotted line). These compounds were detected in samples after single ozonation, exposure to sunlight, and also in the combined process of chlorination with natural sun lightening and artificial lighting supported by NaOCl and UV or O_3_. Elsby et al. [37] indicated that 5-Hydroxybisphenol had a 10-fold less estrogenic potent than BPA. Therefore, the presence of this compound can be harmful to swimming pool users. The presence of the NaOCl chlorination agent leads to the generation of 3,3′-Dichlorobisphenol A, Tetrachlorobisphenol A, 2,4,6-Trichlorophenol, 2,6-Dichlorohydroquinone, 2-Phenylbenzoquinone and Phenol. The first two intermediates were generated by the addition of Cl atoms to the parent compound structure. Other by-products, such as phenols with acid functional groups, alcohol products or benzoquinone are generated by the cleavage of the tertiary carbon-phenolic ring bond [38]. Mutseyekwa et al. [39] reported that the action of O_3_ on BPA lead not only to the formation of single phenolic ring compounds but also to open ring contaminates, such as heptanoic acid and methyl ester.

BPA belongs to one of the most described compounds in the literature. Sharma et al. [36] described the decomposition of this contaminant under HO^•^ and SO_4_^•−^ radical attack and identified nine and seventeen intermediates with a phenolic ring, respectively. Yamamoto and Yasuhara [40] detected eleven chlorination by-products of BPA, and ten different intermediates of the ozonation on this compound were identified by Kusvuran and Yildirim [41]. The interaction of this compound and other phenolic compounds occurring in swimming pool water with several reacting species, however, can therefore lead to the formation of numerous compounds with a potential negative influence on the swimmers.

Three intermediates: the 9-tert-Butyl-3-(2,4-dichloro-5-hydroxyphenyl)-1,3,4-oxadiazol-2(3H)-one, 2,6-Dichlorohydroquinone and 2,4-Dichloropheno were detected in ODZ post-processed samples subjected to the process of ozonation, irradiation with natural sunlight and the combined process of artificial light/chlorine/ozone action. Moreover, 2,4-Dichloropheno is the result of the loos of the nitro group of the 9-tert-Butyl-3-(2,4-dichloro-5-hydroxyphenyl)-1,3,4-oxadiazol-2(3H)-one. Future reactions with highly reactive hydroxyl radicals result in the formation of 2,6-Dichlorohydroquinone. Non by-products were noted for samples after the process of natural sunlight action supported by the presence of NaOCl and O_3_; however, samples after single chlorination contain only 2,6-Dichlorohydroquinone. The conducted analytical methods did not allow for detailed estimation of these CEC degradation pathways. For example, Zhao et al. [42] proposed a decomposition pathway of ODZ during the process of non-thermal plasma treatment. The denitration and dechlorination of the ODZ molecule lead to the formation of several open ring compounds, which completely decompose to CO_2_ and H_2_O by the action of HO^•^ radicals and O_3_. A beneficial impact on the removal degree of O_3_ and the HO^•^ radicals, which were produced during the irradiation of the tested solution with natural sunlight or artificial UV light, was also shown in this study.

It can be concluded that the number of intermediates is strictly related to the removal degree of the parent compound. Post-processed water solutions with a high removal degree of the parent contaminant are characterized by a high number of its intermediates.

## 5. Conclusions

Micropollutants decompose faster when exposed to sunlight than artificial lighting. Therefore, the actual concentrations of these compounds may be higher in indoor swimming pools than in outdoor swimming pools. Some CECs, such as CAF and CBZ, are resistant to light decomposition, therefore their decomposition should be supported by other processes or chemicals which can act as a source of oxidants. The highest CEC removal degrees were noted for the process which combines the action of natural sunlight, NaOCl and O_3_. This process allows also for the decomposition of all CAF and ODZ intermediates. The decomposition by-products are the result of the chlorination and/or oxidation of the parent compounds. The intermediates were mostly identified during the first 30 min of process duration. Between 30 and 60 min of process duration, a gradual decomposition of these compounds was noted. Their chemical structure and also their toxicological nature strictly depend on the implemented decomposition process and the radicals which attract the compound molecule. Their presence in swimming pool water, even in trace concentrations, may pose a negative impact on the health of swimming pool users.

## Figures and Tables

**Figure 1 ijerph-19-05660-f001:**
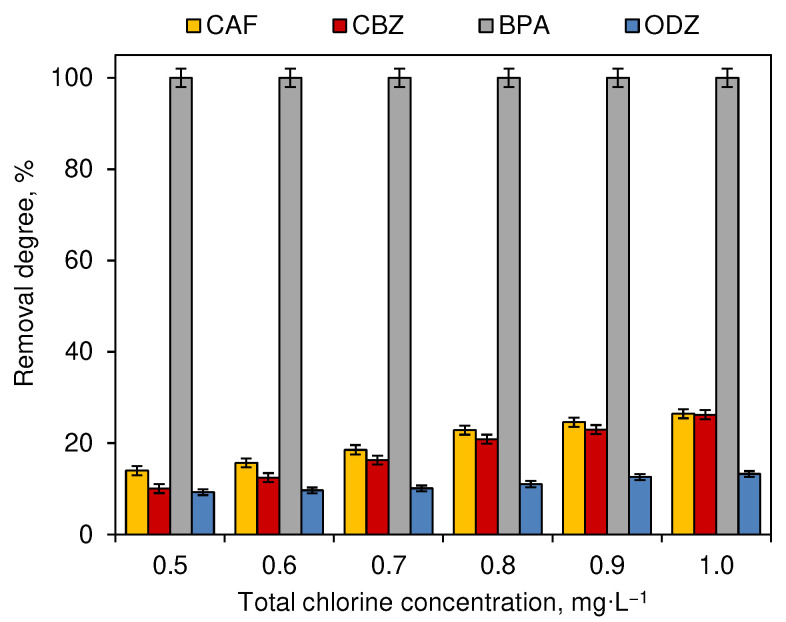
Influence of total chlorine concentration on the removal degree of CECs (dark chamber experiment, *n* = 3 where CAF—caffeine, CBZ—carbamazepine, BPA—bisphenol A, ODZ—oxadiazon).

**Figure 2 ijerph-19-05660-f002:**
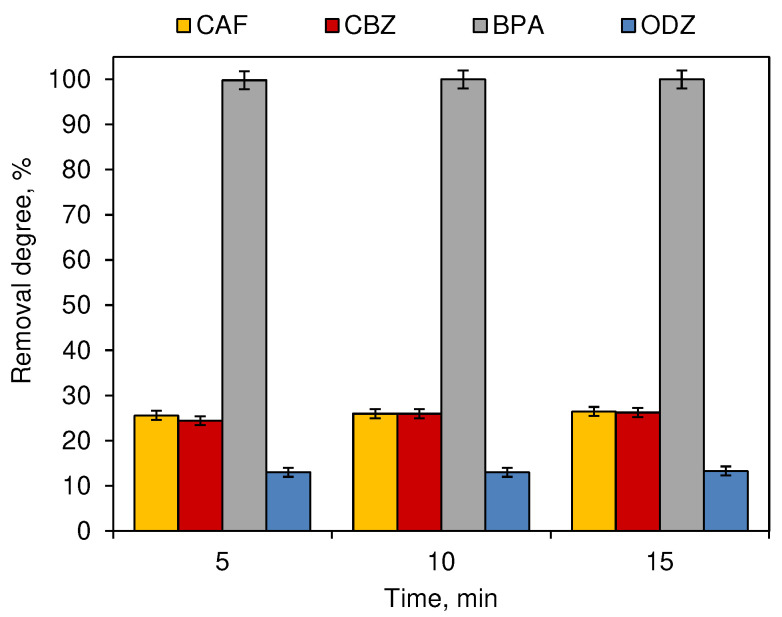
Influence of chlorination time on the removal degree of CECs (dark chamber experiment; *n* = 3; where CAF—caffeine, CBZ—carbamazepine, BPA—bisphenol A, ODZ—oxadiazon).

**Figure 3 ijerph-19-05660-f003:**
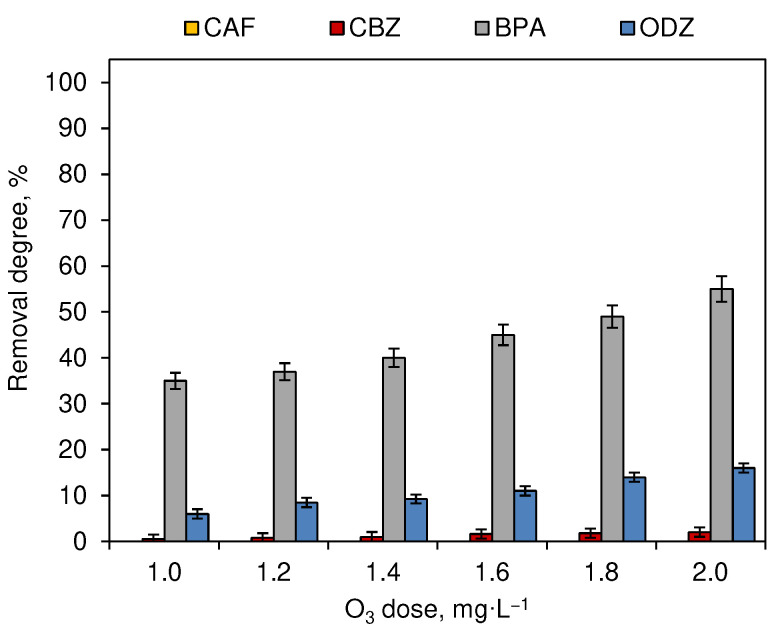
Influence of O_3_ dose on the removal degree of CECs (dark chamber experiment; *n* = 3; where CAF—caffeine, CBZ—carbamazepine, BPA—bisphenol A, ODZ—oxadiazon).

**Figure 4 ijerph-19-05660-f004:**
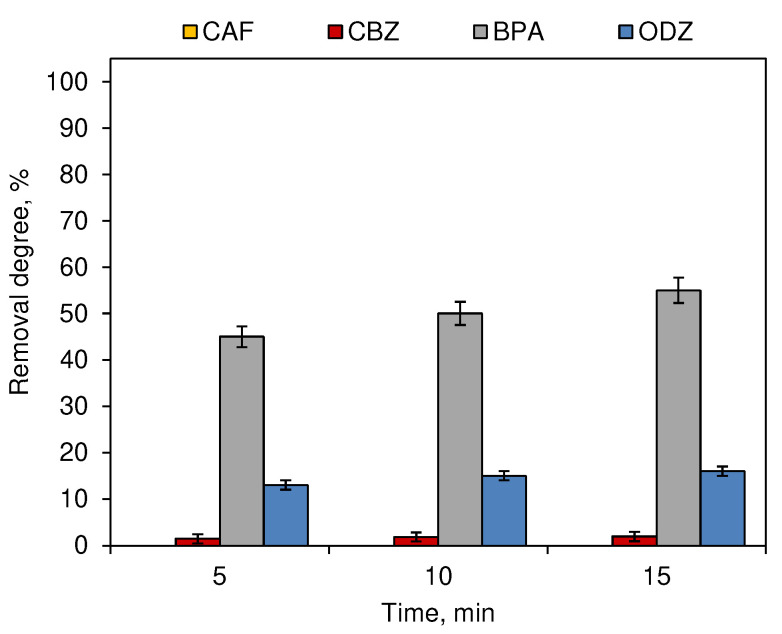
Influence of the O_3_ contact time on the removal degree of CECs (dark chamber experiment; *n* = 3; where CAF—caffeine, CBZ—carbamazepine, BPA—bisphenol A, ODZ—oxadiazon).

**Figure 5 ijerph-19-05660-f005:**
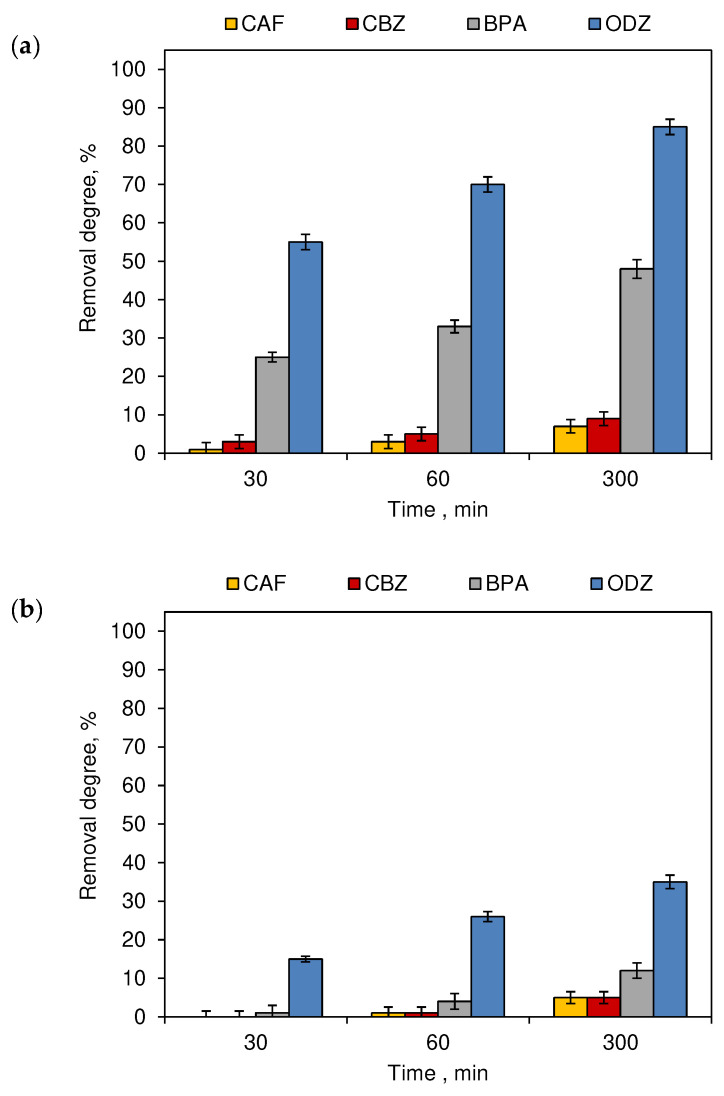
The removal degree of CECs tap water mixtures exposed to (**a**) sunlight (outdoor swimming pool) and (**b**) artificial lighting (indoor swimming pool) (*n* = 3; where CAF—caffeine, CBZ—carbamazepine, BPA—bisphenol A, ODZ—oxadiazon).

**Figure 6 ijerph-19-05660-f006:**
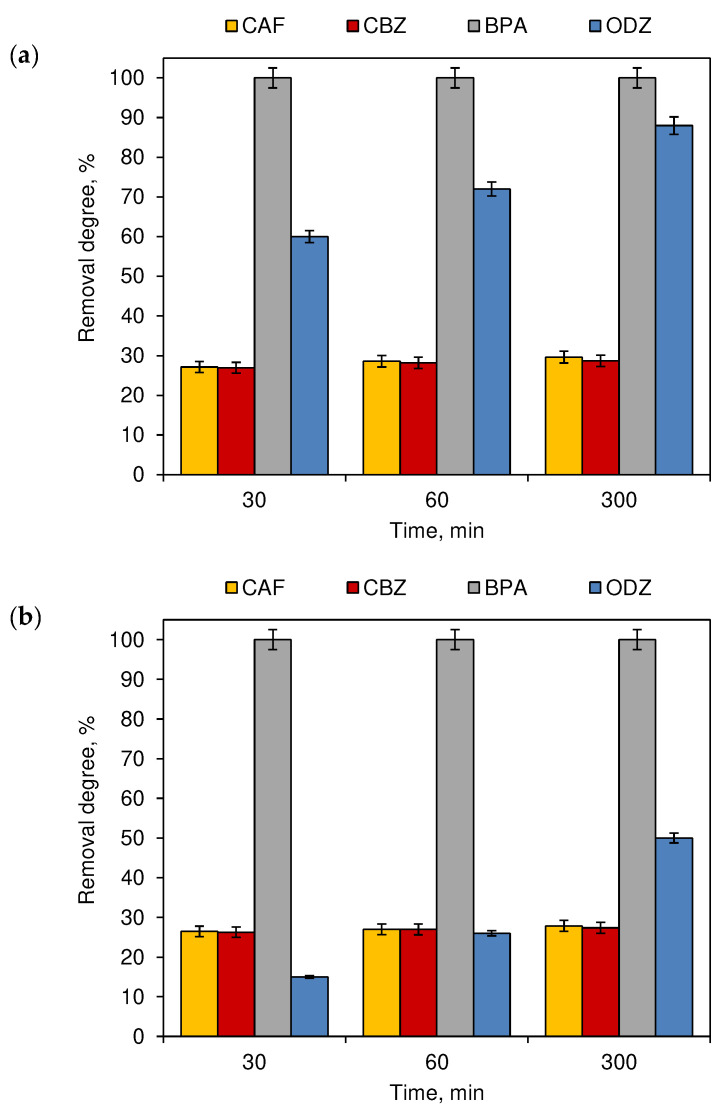
The removal degree of CECs chlorinated swimming pool water mixtures exposed to (**a**) sunlight (outdoor swimming pool) and (**b**) artificial lighting (indoor swimming pool) (*n* = 3; where CAF—caffeine, CBZ—carbamazepine, BPA—bisphenol A, ODZ—oxadiazon).

**Figure 7 ijerph-19-05660-f007:**
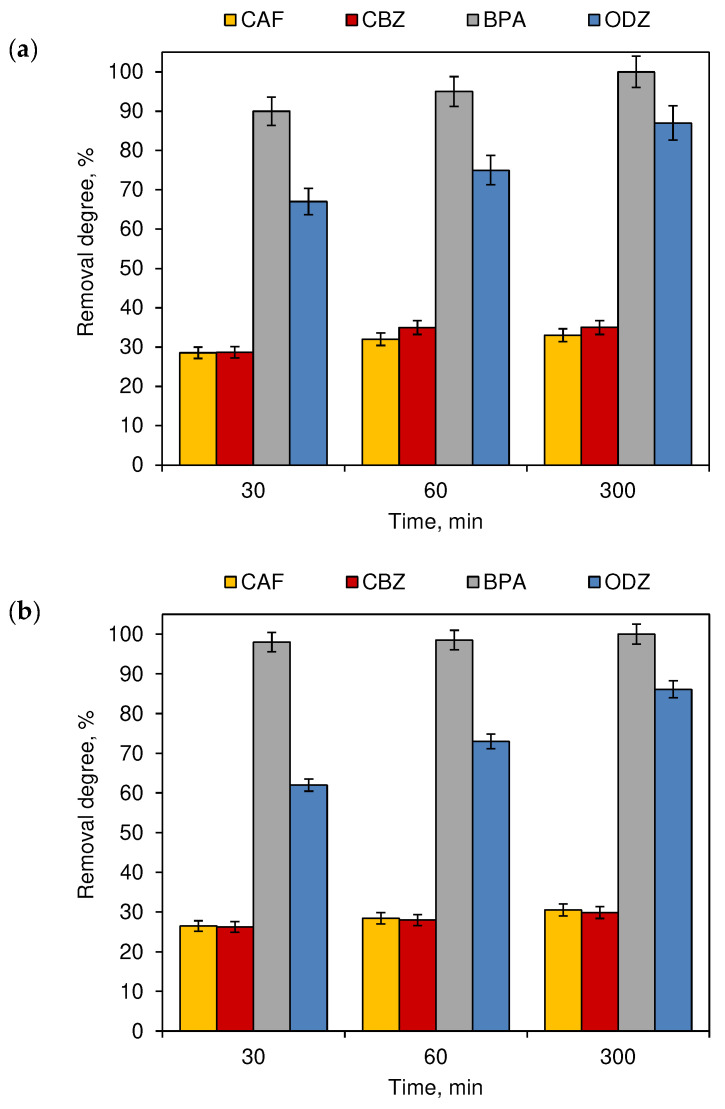
The removal degree of CECs from chlorinated model swimming pool water additionally irradiated by UV light—(**a**) outdoor swimming pool and (**b**) indoor swimming pool (*n* = 3; where CAF—caffeine, CBZ—carbamazepine, BPA—bisphenol A, ODZ—oxadiazon).

**Figure 8 ijerph-19-05660-f008:**
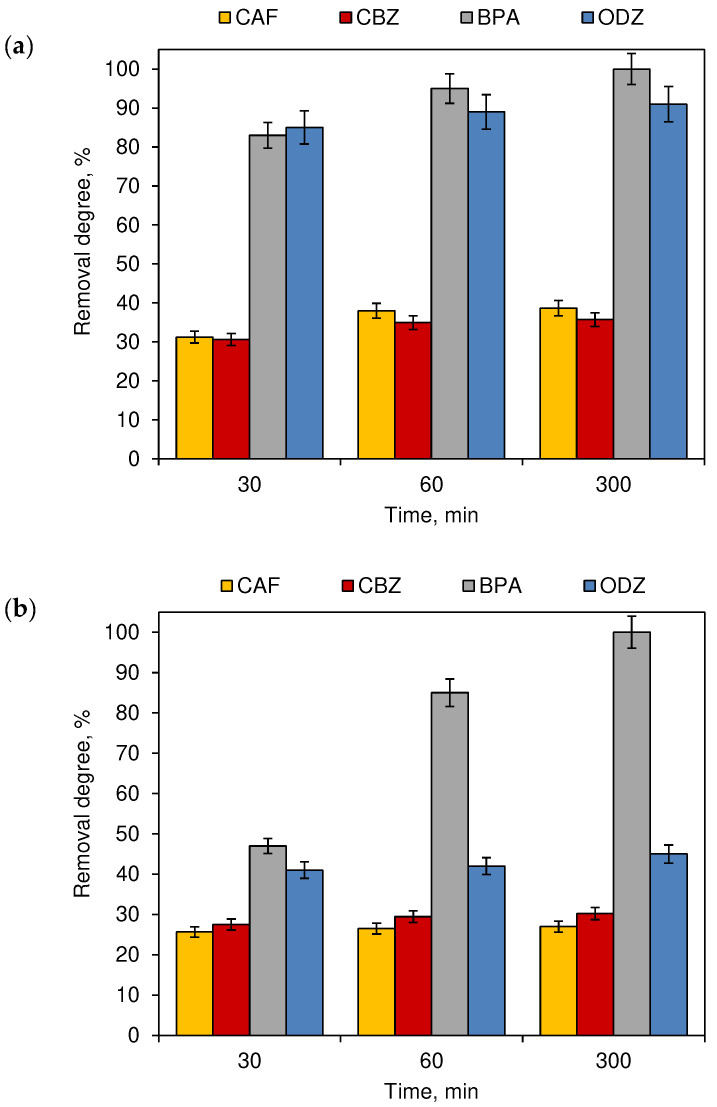
The removal degree of CECs from chlorinated model swimming pool water additionally disinfected by O_3_ light—(**a**) outdoor swimming pool and (**b**) indoor swimming pool (*n* = 3; where CAF—caffeine, CBZ—carbamazepine, BPA—bisphenol A, ODZ—oxadiazon).

**Figure 9 ijerph-19-05660-f009:**
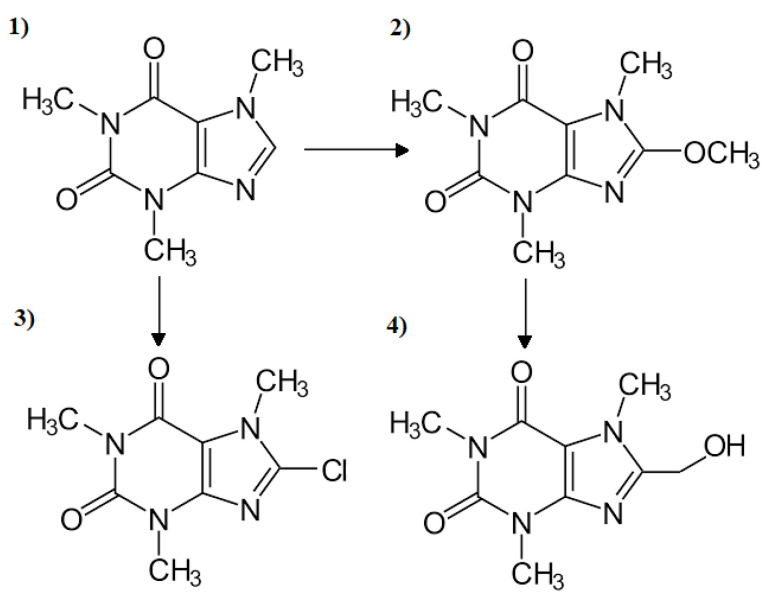
(**1**) CAF decomposition by-products: (**2**) 8-Methoxycaffeine, (**3**) 8-Chlorocaffeine and (**4**) 8-Hydroxymethylcaffeine.

**Figure 10 ijerph-19-05660-f010:**
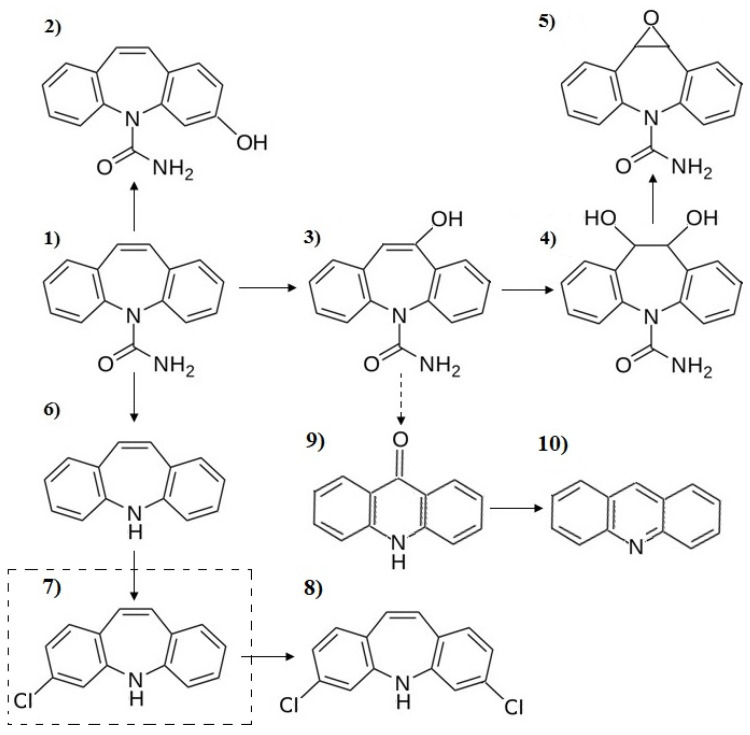
(**1**) CBZ decomposition by-products: (**2**) 3-Hydroxycarbamazepine, (**3**) 10,11-Dihydro-10-hydroxycarbamazepine, (**4**) Dihydrocarbamazepine-10,11-trans-diol, (**5**) Carbamazepine-10,11-epoxide, (**6**) Iminostilbene, (**7**) MW = 227 3-Chloro-5H-dibenz[b,f]azepine, (**8**) 3,7-dichloro-5h-dibenz[b,f]azepine, (**9**) 9-Acridone and (**10**) Acridine.

**Figure 11 ijerph-19-05660-f011:**
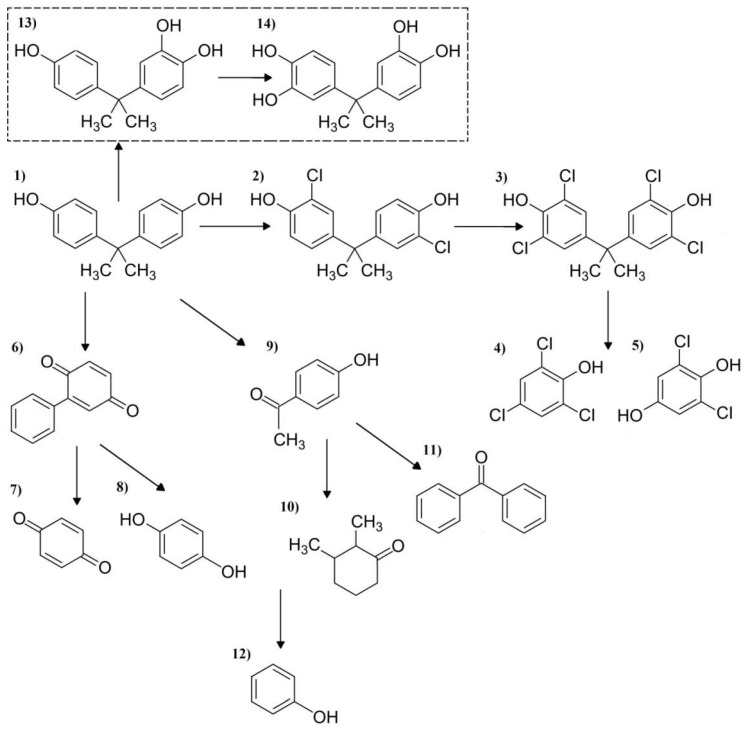
(**1**) BPA decomposition by-products: (**2**) 3,3′-Dichlorobisphenol A, (**3**) Tetrachlorobisphenol A, (**4**) 2,4,6-Trichlorophenol, (**5**) 2,6-Dichlorohydroquinone, (**6**) 2-Phenylbenzoquinone, (**7**) Benzoquinone, (**8**) Hydroquinone, (**9**) *p*-Hydroxyacetophenone, (**10**) 2,3-Dimethylcyclohexanon, (**11**) Benzophenone, (**12**) Phenol, (**13**) MW = 244 5-Hydroxybisphenol and (**14**) MW = 262 5,5′-Dihydroxybisphenol.

**Figure 12 ijerph-19-05660-f012:**
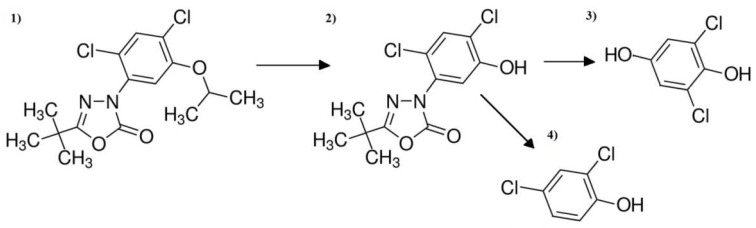
(**1**) ODZ decomposition by-products: (**2**) 9-tert-Butyl-3-(2,4-dichloro-5-hydroxyphenyl)-1,3,4-oxadiazol-2(3H)-one, (**3**) 2,6-Dichlorohydroquinone and (**4**) 2,4-Dichloropheno.

**Table 1 ijerph-19-05660-t001:** Characteristics of the model swimming pool water.

Parameter	Unit	Value
pH	-	7.2
Conductivity	µS∙cm^−1^	972.6
Temperature	°C	28
TOC *	mg·L^−1^	3.23
IC **	mg·L^−1^	1.10
TC ***	mg·L^−1^	4.33
Free chlorine	mg·L^−1^	0.5
Combined chlorine	mg·L^−1^	0.1
Total chlorine	mg·L^−1^	0.6
Turbidity	NTU	0.1

* TOC—Total organic carbon, ** IC—inorganic carbon, *** TC—Total carbon.

**Table 2 ijerph-19-05660-t002:** Occurrence of by-products in the post-processed water samples.

Compound	Processes									
	Cl_2_ *	O_3_	NL	AL	NL/Cl_2_	AL/Cl_2_	NL/Cl_2_/UV	AL/Cl_2_/UV	NL/Cl_2_/O_3_	AL/Cl_2_/O_3_
CAF by-products										
8-Methoxycaffeine	-	-	+	+	+	+	-	+	-	+
8-Chlorocaffeine	+	-	-	-	+	+	-	+	-	+
8-Hydroxymethylcaffeine	-	-	+	+	+	+	-	+	-	+
CBZ by-products										
3-Hydroxycarbamazepine	+	+	+	+	+	+	+	+	+	+
10,11-Dihydro-10-hydroxycarbamazepine	+	-	+	+	+	+	+	+	+	+
Dihydrocarbamazepine-10,11-trans-diol	+	-	+	-	+	+	+	+	+	+
Carbamazepine-10,11-epoxide	+	+	+	-	+	+	+	+	-	+
Iminostilbene	+	-	+	-	+	-	+	+	-	+
3-Chloro-5H-dibenz[b,f]azepine	+	-	-	-	+	+	+	+	+	+
3,7-dichloro-5h-dibenz[b,f]azepine	+	-	-	-	+	+	+	+	+	+
9-Acridone	-	-	+	-	+	-	+	-	-	-
Acridine	-	-	+	-	+	-	+	-	+	+
BPA by-products										
3,3′-Dichlorobisphenol A	+	-	-	-	+	+	+	+	-	+
Tetrachlorobisphenol A	+	-	-	-	+	+	+	+	-	+
2,4,6-Trichlorophenol	+	-	-	-	+	+	+	+	+	+
2,6-Dichlorohydroquinone,	+	-	-	-	+	+	+	+	+	+
2-Phenylbenzoquinone	+	+	+	-	+	+	+	+	+	+
Benzoquinone	-	+	+	-	+	+	+	+	+	+
Hydroquinone	-	+	+	+	+	-	+	+	+	+
p-Hydroxyacetophenone	-	+	+	+	+	+	+	+	+	+
2,3-Dimethylcyclohexanon	-	+	+	+	+	+	+	+	+	+
Benzophenone	-	+	+	-	+		+		+	+
Phenol	+	+	+	+	+	+	+	+	+	+
5-Hydroxybisphenol	-	+	+	+	+	-	-	+	-	+
5,5′-Dihydroxybisphenol	-	+	+	-	+	-	-	+	-	+
ODZ by-products										
9-tert-Butyl-3-(2,4-dichloro-5-hydroxyphenyl)-1,3,4-oxadiazol-2(3H)-one	-	+	+	+	+	+	+	+	-	+
2,6-Dichlorohydroquinone	+	+	+	-	+	+	+	+	-	+
2,4-Dichloropheno	-	+	+	-	-	-	-	-	-	+

* Cl_2_—chlorination, O_3_—ozonation, NL—natural sunlight, Al—artificial light, UV—UV radiation.

## Data Availability

Not applicable.

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
