# Peer review of "Identification of Potential Harmful Transformation Products of Selected Micropollutants in Outdoor and Indoor Swimming Pool Water"

_ijerph, 2022, doi:10.3390/ijerph19095660_

Round 1

Reviewer 1 Report

This manuscript deals with a very important topic as the emerging contaminants are of global concern. But, several points need to be addressed to fit for publication as follows:
1.    A major concern in this manuscript is that the detail of the statistical analysis of the data is missed. Thus, a separate section of the statistical analysis should be added to clarify every detail of the statistical analysis. In addition, in the results section, the data should be described as significant or non-significant and the P-value of significance should be added. 
2.    It is not recommended to begin sentences with abbreviations like “NaClO” in line 75 and “BPA” in line 395.
3.    Table 1: the full term of all abbreviations used within the table should be clarified in the footnote.
4.    In figures 1, 2, 6, 7, and 8, there are some means that exceeded 100 thus the Y-axis should be extended to 110 or 120.
5.    In the figure legends: the full term of all abbreviations used should be clarified. Also, n=? should be mentioned and clarified if the means were presented as means ± SE or SD.
6.    Merging of the results and discussion in one section make the interpretations of the findings of the study unclear. Thus, it is highly recommended to separate results and discussion and give more detailed interpretations of the findings of the study.

Author Response

We would like to thank all Reviewers for their valuable help in improving our manuscript. As can be seen in the revised text, their comments, remarks, and suggestions have been taken into account during our revision. Detailed responses to reviewers are summarized below, and the changes are marked in red in the uploaded revised version of the manuscript.

Reviewer: 1

This manuscript deals with a very important topic as the emerging contaminants are of global concern. But, several points need to be addressed to fit for publication as follows:
1.    A major concern in this manuscript is that the detail of the statistical analysis of the data is missed. Thus, a separate section of the statistical analysis should be added to clarify every detail of the statistical analysis. In addition, in the results section, the data should be described as significant or non-significant and the P-value of significance should be added. 

Thank you very much for your extensive review and accurate comments. Your remarks allowed us to improve the quality of our work. We supplemented the manuscript with the statistical analysis details. The detailed description was given in subchapter 2.5. Evaluation of research results.

  1.  It is not recommended to begin sentences with abbreviations like “NaClO” in line 75 and “BPA” in line 395.

Thank you very much for your advice. We correct this mistake.

  1. Table 1: the full term of all abbreviations used within the table should be clarified in the footnote.

Thank you for your advice. We supplemented the table description in the footnote.

  1.  In figures 1, 2, 6, 7, and 8, there are some means that exceeded 100 thus the Y-axis should be extended to 110 or 120.

Thank you very much for your advice. We correct the Y-axis.

  1.  In the figure legends: the full term of all abbreviations used should be clarified. Also, n=? should be mentioned and clarified if the means were presented as means ± SE or SD.

Thank you for your advice. We supplemented the figure description.

  1. Merging of the results and discussion in one section make the interpretations of the findings of the study unclear. Thus, it is highly recommended to separate results and discussion and give more detailed interpretations of the findings of the study.

Thank you for your comment. The manuscript was rewritten and the results and discussion was presented in two separate chapters.

Reviewer 2 Report

This article is a fascinating study on how different pool water treatments can influence indoor and outdoor swimming pool water decontamination. The main strength of the work is the proposal of the descomposition of pathways which adds extra value to the paper. However, the authors should consider further discussion of the results obtained to improve. Additionally, to be published in the IJERPH journal, authors should consider certain aspects of the text:

  • Point 2.3: The authors say that the assay was performed between 24-26 ºC. Was the temperature controlled? Perhaps it would be advisable to indicate this in the text.
  • Point 2.4: The title does not describe the content (is the same as section 2.3). Please check.
  • The sentence in lines 140-143 is not clear enough. Please clarify.
  • In line 146, the authors use the acronym MeOH to refer to methanol. However, this could confuse a reader who is not very familiar with the abbreviations used in chemistry.
  • Results or discussion: Although the results are described with sufficient clarity, my main concern is the lack of discussion of the results and comparison with other published work. I do not know if this may be due to the lack of previous work, but it should be specified if so. Otherwise, the authors should compare/discuss their results with previous work.
  • Line 195: The authors may refer to figure 3 when talking about figure 1. Please check.
  • Figure 6: Only one figure appears for the 300 minutes. Why do not two graphs compare indoor and outdoor swimming pools as the other CECs?
  • Lines 331-332: The authors make a somewhat risky claim. I would ask the authors to reconsider that statement.
  • Conclusión: In general, the conclusion meets expectations. Perhaps, to improve it, the authors should think about introducing a paragraph referring to the last point of the discussion about the pathways of decomposition.

Additionally, some spelling mistakes have been detected. Therefore, I recommended authors re-read the whole text, reviewing this issue.

Author Response

We would like to thank all Reviewers for their valuable help in improving our manuscript. As can be seen in the revised text, their comments, remarks, and suggestions have been taken into account during our revision. Detailed responses to reviewers are summarized below, and the changes are marked in red in the uploaded revised version of the manuscript.

Reviewer: 2

This article is a fascinating study on how different pool water treatments can influence indoor and outdoor swimming pool water decontamination. The main strength of the work is the proposal of the descomposition of pathways which adds extra value to the paper. However, the authors should consider further discussion of the results obtained to improve. Additionally, to be published in the IJERPH journal, authors should consider certain aspects of the text:

  • Point 2.3: The authors say that the assay was performed between 24-26 ºC. Was the temperature controlled? Perhaps it would be advisable to indicate this in the text.

Thank you for your comment. The temperature of the tested solutions was continuously measured during the experiments by the use of the multifunctional meter CX-461 from ELMETRON (Zabrze, Poland). This information was added to the manuscript.

  • Point 2.4: The title does not describe the content (is the same as section 2.3). Please check.

Thank you for your remark. The title of the subchapter 2.4 was corrected as follows:

2.4. Compound analytical procedure

  • The sentence in lines 140-143 is not clear enough. Please clarify.

Thank you for your remark. The sentences were rewritten as follows:

“Supelclean™ ENVI-18 by Sigma-Aldrich (Poznań, Poland) were used as extraction columns. The column bed of those type of cartridges was composed of silica gel base material with polymericallyoctadecyl bonding (bed weight of 1000 mg). The bed pore size was equal to 60 Å and the total bed surface area reached 475 m2 g-1.”

  • In line 146, the authors use the acronym MeOH to refer to methanol. However, this could confuse a reader who is not very familiar with the abbreviations used in chemistry.

Thank you for your comment. We replace the acronyms with full names of the used solvents.

  • Results or discussion: Although the results are described with sufficient clarity, my main concern is the lack of discussion of the results and comparison with other published work. I do not know if this may be due to the lack of previous work, but it should be specified if so. Otherwise, the authors should compare/discuss their results with previous work.

Thank you for your comment. The manuscript was rewritten and the results and discussion were presented in two separate chapters.

  • Line 195: The authors may refer to figure 3 when talking about figure 1. Please check.

Thank you for your remark. We correct this mistake.

  • Figure 6: Only one figure appears for the 300 minutes. Why do not two graphs compare indoor and outdoor swimming pools as the other CECs?

Thank you for your comment. The data presented on this figure was supplemented by the results obtained after 30 and 60 min of process duration.

  • Lines 331-332: The authors make a somewhat risky claim. I would ask the authors to reconsider that statement.

Thank you for your comment. Our statement was supported by literature data:
“This indicated the presence of a Cl in the chemical structure of the compound. It can be supposed that this compound was generated during the chlorination of Iminostilbene (Fig. 10). The Cl atom attached the phenol ring and 3-Chloro-5H-dibenz[b,f]azepine was generated, which after the connection with a second Cl atom transforms into 3,7-dichloro-5h-dibenz[b,f]azepine. This possibility was also noted in previous studies focused on the chlorination of CBZ. Tandarić et al. [1] pointed that Iminostilbene can undergo different transformation reactions promoted by reactive chlorine species. However the two most feasible processes can be the additions of -Cl or -OH group on the C=C bond of this compound.”

  1. Tandarić, T.; Vrček, V.; Šakić, D. A quantum chemical study of HOCl-induced transformations of carbamazepine. Org BiomolChem. 2016, 14,10866-10874
  • Conclusión: In general, the conclusion meets expectations. Perhaps, to improve it, the authors should think about introducing a paragraph referring to the last point of the discussion about the pathways of decomposition.

Thank you for your remark. The conclusions were supplemented with the information about the pathways of decomposition.

Additionally, some spelling mistakes have been detected. Therefore, I recommended authors re-read the whole text, reviewing this issue.

Thank you for your remark.We carefully re-read the whole text and correct the spelling mistakes.

Reviewer 3 Report

A relevant manuscript on a topic that is important for evaluating human health impacts and optimal water treatment technologies. Specific comments include the following:

Lines 64-66: Whereas CAF can be excreted in urine and BPA could coat water supply pipes, CBZ is an anti-convulsant medication and ODZ an herbicide—are there any references for how these latter two compounds are routinely introduced to swimming pools?

Lines 97-102: A TOC of 3.23 mg/L is quite low for a swimming pool—is there a reason you selected this value or was it the concentration in tap water? And is the tap water in your lab not chlorinated?

Line 146: What is CAN—do you mean ACN? I assume that ACN is an acronym for acetonitrile; however, that is not explained in the text.

Line 172: Does the term “removal degrees” refer to removal efficiencies?

Line 185: Does the term “elongation” refer to lengthening or increasing the time of that you allowed chlorination reactions to occur?

Lines 195: I think your reference to Figure 1 should actually be Figure 3.

Lines 213-217: The greater removal of ODZ by ozonation, compared to chlorination, is only marginal. Also, you might expand on which physical and chemical properties of the four compounds render them more or less susceptible to the two disinfectants.

Lines 284-288: These 3 sentences don’t seem to follow from the results you presented. Consider re-wording this observation in a manner that is more understandable to readers.

Lines 349-352: At a pH of 7.2, would you not expect HOCl to be the major reactive species of NaOCl? If not, what “other” chlorine species would you expect to be major contributors to the chlorination reactions?

Figures 10-12: With regard to your potential pathways for the formation of chemical intermediates, do you have any information regarding the kinetics that would indicate which ones are likely to persist and which are shorter-term intermediates? Was the time interval over which you allowed the reactions to proceed the same as those you used to document removal efficiencies of the parent compounds? Including this information in the figure captions would be helpful.

Author Response

We would like to thank all Reviewers for their valuable help in improving our manuscript. As can be seen in the revised text, their comments, remarks, and suggestions have been taken into account during our revision. Detailed responses to reviewers are summarized below, and the changes are marked in red in the uploaded revised version of the manuscript.

Reviewer: 3

A relevant manuscript on a topic that is important for evaluating human health impacts and optimal water treatment technologies. Specific comments include the following:

Lines 64-66: Whereas CAF can be excreted in urine and BPA could coat water supply pipes, CBZ is an anti-convulsant medication and ODZ an herbicide—are there any references for how these latter two compounds are routinely introduced to swimming pools?

Thank you for your comment. The last chapter of the Introduction was rewritten and supplemented with information about the studied micropollutants.

Lines 97-102: A TOC of 3.23 mg/L is quite low for a swimming pool—is there a reason you selected this value or was it the concentration in tap water? And is the tap water in your lab not chlorinated?

Thank you for your remarks and questions. During the realization of the project No. 2018/29/N/ST/01352 (this study is a part of it), we analysed swimming pool water form more than 50 different pool located in Poland. The TOC of 3.23 mg/L is typical for this type of water in a properly functioning pool water treatment system.

The average TOC of our tap water is 1.45 mg/L. The tap water is chlorinated, but we always measure the actual chlorine concentration and we set the dose of introduced NaOCl so that the test water had the desired concentration of total chlorine.

Line 146: What is CAN—do you mean ACN? I assume that ACN is an acronym for acetonitrile; however, that is not explained in the text.

Thank you for noticing the typographical mistake. We replace the acronyms with full names of the used solvents.

Line 172: Does the term “removal degrees” refer to removal efficiencies?

Thank you for your remark. The “removal degrees” refer to removal efficiencies. We correct the description in the main text of the manuscript.

Line 185: Does the term “elongation” refer to lengthening or increasing the time of that you allowed chlorination reactions to occur?

Thank you for your remark. We change the term “elongation” to “increasing time”.

Lines 195: I think your reference to Figure 1 should actually be Figure 3.

Thank you for your remark. We correct this mistake.

Lines 213-217: The greater removal of ODZ by ozonation, compared to chlorination, is only marginal. Also, you might expand on which physical and chemical properties of the four compounds render them more or less susceptible to the two disinfectants.

Thank you for your comment. The manuscript was rewritten and the results and discussion were presented in two separate chapters.

Lines 284-288: These 3 sentences don’t seem to follow from the results you presented. Consider re-wording this observation in a manner that is more understandable to readers.

Thank you for your comment. The manuscript was rewritten and the results and discussion were presented in two separate chapters.

Lines 349-352: At a pH of 7.2, would you not expect HOCl to be the major reactive species of NaOCl? If not, what “other” chlorine species would you expect to be major contributors to the chlorination reactions?

Thank you for your comment. We completely agree with you. The sentences were rewritten as follows: “In general the disinfection of swimming pool water by the use of NaOCl lead to the formation of several chlorinated intermediates: HOCl (which is the major chlorinating species), OCl-, ClOH•-, Cland Cl2•-.”

Figures 10-12: With regard to your potential pathways for the formation of chemical intermediates, do you have any information regarding the kinetics that would indicate which ones are likely to persist and which are shorter-term intermediates? Was the time interval over which you allowed the reactions to proceed the same as those you used to document removal efficiencies of the parent compounds? Including this information in the figure captions would be helpful.

Thank you for your comment. The following explanation was added to the text:

“The intermediated were mostly identified during the first 30 minutes of process duration. Signal intensity increased in the time from 0 to 30 minutes, and then between 30 and 60 minutes, the decrease in signal intensity was noted, which may indicate a gradual decomposition of these compounds. In addition, no presence of these compounds was recorded after 300 minutes (especially in the case of post-processed water samples of NL/Cl2/UV, AL/Cl2/UV, NL/Cl2/O3, AL/Cl2/O3).”

Round 2

Reviewer 1 Report

No further comments to be addressed

Reviewer 2 Report

I want to congratulate the authors for the effort to revise the article, taking into account the comments provided. From my point of view, the new version is a significant improvement on the previous one, which makes it suitable for publication in IJERPH if the editor considers it appropriate.